# A Multi-Omics Overview of Colorectal Cancer to Address Mechanisms of Disease, Metastasis, Patient Disparities and Outcomes

**DOI:** 10.3390/cancers15112934

**Published:** 2023-05-26

**Authors:** Guang Yang, Xi (Richard) Yu, Daniel J. Weisenberger, Tao Lu, Gangning Liang

**Affiliations:** 1School of Sciences, China Pharmaceutical University, Nanjing 211121, China; yangguang5@cgeinc.com; 2China Grand Enterprises, Beijing 100101, China; yuxi@eastchinapharm.com; 3Huadong Medicine Co., Ltd., Hangzhou 310011, China; 4Department of Biochemistry and Molecular Medicine, Keck School of Medicine, University of Southern California, Los Angeles, CA 90033, USA; dan.weisenberger@med.usc.edu; 5USC Institute of Urology, Keck School of Medicine, University of Southern California, Los Angeles, CA 90033, USA; 6State Key Laboratory of Natural Sciences, China Pharmaceutical University, Nanjing 211121, China; 7USC Norris Comprehensive Cancer Center, Keck School of Medicine, University of Southern California, Los Angeles, CA 90033, USA

**Keywords:** colorectal cancer, DNA methylation, CIMP, disparities, metastases

## Abstract

**Simple Summary:**

Human colorectal cancer (CRC) is a global health burden, most notably for minority populations. While screening tools have helped improve disease detection and surveillance, CRC is a complex disease with a diverse set of molecular features that are linked to the location of the primary tumor. These features provide challenges to treatment and improving patient outcome. In addition, tumor metastases to liver and other organ systems are a main cause of CRC-related mortality and represent a substantial obstacle to improving patient outcomes. In this review, we summarize CRC tumor subgroups, their molecular features, treatments and outcomes with attention to health disparities and tumor metastases.

**Abstract:**

Human colorectal cancer (CRC) is one of the most common malignancies in men and women across the globe, albeit CRC incidence and mortality shows a substantial racial and ethnic disparity, with the highest burden in African American patients. Even with effective screening tools such as colonoscopy and diagnostic detection assays, CRC remains a substantial health burden. In addition, primary tumors located in the proximal (right) or distal (left) sides of the colorectum have been shown to be unique tumor types that require unique treatment schema. Distal metastases in the liver and other organ systems are the major causes of mortality in CRC patients. Characterizing genomic, epigenomic, transcriptomic and proteomic (multi-omics) alterations has led to a better understanding of primary tumor biology, resulting in targeted therapeutic advancements. In this regard, molecular-based CRC subgroups have been developed that show correlations with patient outcomes. Molecular characterization of CRC metastases has highlighted similarities and differences between metastases and primary tumors; however, our understanding as to how to improve patient outcomes based on metastasis biology is lagging and remains a major obstacle to improving CRC patient outcomes. In this review, we will summarize the multi-omics features of primary CRC tumors and their metastases across racial and ethnic groups, the differences in proximal and distal tumor biology, molecular-based CRC subgroups, treatment strategies and challenges for improving patient outcomes.

## 1. Introduction

Human colon and rectal cancers are worldwide health burdens, with over 1.9 million new combined cases and 935,000 deaths in 2020, accounting for approximately 9–10% of all cancer incidence and mortality [1]. Colorectal cancer (CRC) is the third most diagnosed cancer and the second leading cause of cancer death [1], with an overall balance in numbers between males and females. Globally, CRC incidence is highest in Australia, Europe and the USA but lower in underdeveloped nations. CRC incidence is linked to diet/obesity, smoking, alcohol and lack of exercise, but genetic factors and family history are also influential in determining disease risk. In recent decades, CRC incidence rates for adults over 50 have decreased, mainly due to improved screening methods such as colonoscopy, imaging, and non-invasive biomarker tests. However, early-onset disease incidence for adults under 45 years old has increased in developed nations [1]. CRC mortality is mainly driven by metastases to distal tissues, mainly liver and brain tissues. Indeed, CRC metastases to liver tissues occur in 33–50% of all CRC patients [2,3] and reduce the overall survival (OS) to 3–11% if left untreated or given palliative care [2,4,5]. However, OS rates can climb up to 60% if the liver metastases are treated by removal of the affected liver tissue and thermal ablation of the tumor tissue [6,7].

## 2. Racial Disparities in CRC

As with every form of human cancer, racial disparities in incidence and outcome are present for CRC patients. African Americans followed by American Indians display increased CRC incidence, advanced disease stage and mortality as compared to Caucasian Americans [8]. African American patient CRC incidence is generally 3–7 years earlier than for Caucasian patients. African Americans are less likely to know their own family history of disease, to relay the detection of colorectal polyps to family members and in general, are distrustful of the US medical system [9]. The incidence disparity occurs even for early-onset patients under 45 years old. These findings suggest that several factors limit access to health care [8], including awareness and knowledge of recommended screenings; socioeconomics; healthcare access; limited, inadequate or no health insurance; or other barriers that ultimately limit preventative screening, treatment and follow-up health care [10,11,12,13]. From a socioeconomic perspective, individuals in lower socioeconomic populations also show lower participation in disease screening worldwide [14]. In general, racial disparities in patient outcomes are mainly related to disparities in CRC screening and clinical access and treatment differences [11].

With respect to treatment disparities, one report showed that while African American and Caucasian CRC patients were examined by oncologists at similar or equal rates, Caucasian patients were more likely to receive chemotherapy [11,15]. However, this is confounded by a recent study [16] showing that among stage III CC patients, a higher proportion of African American patients began adjuvant chemotherapy compared to Caucasians. It should be noted that duration, type, commitment and intensity of treatments were not included in the analyses and may be contributing factors to disparity [17]. African Americans have higher rates of recurrence and lower rates of successful therapy, further complicating the true sources of CRC health disparities [17].

Treatment disparities are also related to those physicians treating African American versus Caucasian patients, such that medical personnel treating African American patients are more likely to be trained outside of the United States and are less familiar with current screening methods, use older technologies, are understaffed, have overall less funding and have overall higher patient volumes. Since minority groups are generally geographically segregated, the access to local quality care is more limited, thereby resulting in critical overdemand of medical access that is inherently compromised [11]. Indeed, a report from Obrochta and colleagues showed that minority CRC patients were at an increased risk for delayed treatment or undertreatment compared to Caucasian patients, which is derived from neighborhood socioeconomic status [18].

A study led by Warren Anderson [19] investigated sociocultural factors, as well as environmental/dietary exposures and healthcare access among Caucasian and African American CRC patients and showed that increased screening and higher patient education were associated with lower CRC risk among African Americans [19,20]. In addition, CRC incidence disparity is not evident when correcting for screening, suggesting that screening is a key driver of CRC health disparities. In support of this, CRC incidence disparities were not explained fully by family history, healthcare access, environmental/dietary exposures or socioeconomic background [19,20].

Similar conclusions were reached by Kane et al., who showed that while all minority groups showed lower CRC screening rates compared to Caucasians, this disparity was reduced after adjustment for socioeconomic and behavioral factors [21]. Improved screening of both African Americans and Caucasians within a healthcare system over a 20-year period from 2000–2019 [22] and data from the USA National Cancer Institute (reviewed in [23]) have resulted in overall lower CRC incidence, as well as lower incidence of early- and late-stage disease and CRC mortality. Even though the incidence and death rates of African Americans are still higher than those of Caucasian patients, this shows that earlier detection and treatment are beneficial to improved patient outcomes across racial groups.

Early-onset CRC is becoming increasingly prevalent in the US population over the past 20 years, such that the recommended age for colonoscopy-based screening for polyps, adenomas and adenocarcinomas is now 45 years of age for the general population [24,25]. Early-onset CRC patients are more likely to present with advanced stage (stage IV) cancer and to be African American or Hispanic [26]. Specifically, early-onset African American CRC patients display worse OS compared to Caucasian patients across all income levels, even those patients who are more educated, have a higher income and have health insurance [24]. Petrick and colleagues also showed an increase in early-onset CRC among minority populations compared to Caucasian patients. Late-onset cases are most common in men, with the location in the left side of the colon, while women present with right-sided disease [27]. Interestingly, incidence of early-onset CRC was more prevalent in women [27]. Generally, CRCs are predominantly located in the distal (left) side of the colorectum, and African American patients show a bias towards primary tumors located in the proximal (right) area of the colon with microsatellite stable (MSS) disease. Interestingly, early-onset CRC patients present with left-sided primary tumors that have aggressive behavior [26], suggesting that there are also biological factors that help drive CRC disparities between racial and ethnic groups.

## 3. Genetic Progression of CRC

CRC progresses from the development of precancerous polyps from colonic epithelium to the formation of adenomas, followed by the development of adenocarcinomas that may ultimately invade the colonic epithelium and metastasize to liver and other tissues. Most CRCs develop sporadically, while only 5% of CRCs are derived from germline transmission of mutated driver genes. Among the heritable CRCs, Lynch syndrome represents less than 4% of all CRCs and occurs due to germline mutations in several mismatch repair genes, including *MLH1*, *MSH2* and *MSH6* [28].

CRC has been well studied with respect to the molecular alterations that drive tumor development and progression. Vogelstein originally proposed driver somatic mutation and deletion events that are key in the adenoma-carcinoma pathway that align with tumor cell selection and clonal expansion [29]. In this model, *APC* alterations (mutations or deletions) represent the first events in CRC tumorigenesis as normal colonic epithelial cells convert into hyperproliferative cells. APC alterations occur in up to 90% of all CRCs [30,31] and result in sustained Wnt signaling through stabilized β-catenin expression. Activating *KRAS* mutations (*KRAS*-mut), chromosome 18 deletions that include the *DCC* locus, and inactivation of the TGF-β response by *SMAD2*/*SMAD4* changes are key steps in the development of adenomas. Finally, *TP53* alterations (mutations and/or deletions) represent the gatekeeper events in the development of adenocarcinomas (Figure 1).

More recently, the Big Bang model [32] was developed and argues that tumors grow as a single expansion after an initial cellular transformation. The Big Bang model is supported by the fact that advanced tumors do not generally display clonal selection. When applied to metastases, the Vogelstein model assumes linearity in a stepwise fashion through novel mutations and their expansion by clonal selection [29]. Alternatively, the Big Bang model suggests that the metastatic potential of a tumor cell is defined early during tumorigenesis [32,33], in which a metastatic cell is selected based on growth advantage. In support of this, Hu and colleagues [33] performed WES on pairs of CRCs and liver metastases to show that driver mutations are formed early and that the mutation profiles between CRCs and their liver metastasis pair show strong convergence. In addition, metastases are seeded when the tumor is undetectable.

While approximately two thirds of CRCs develop from the traditional adenoma pathway, one third of CRCs develop from the serrated pathway [34,35]. These include hyperplastic polyps (HPPs), sessile serrated adenomas (SSAs) and traditional serrated adenomas (TSAs). HPPs are mainly located in the distal (left) side of the colon but do not develop past adenomas. TSAs are low-frequency serrated polyps that present in the distal colon with a sawtooth-like appearance and are enriched for *KRAS* mutations. SSAs are mainly located in the proximal (right) colon, display the *BRAF (V600E)* point mutation (BRAF-mut) and are underscored by increased size with a flattened and serrated appearance [34,35] (Figure 2).

These findings establish differences in tumor etiology and molecular features between CRCs that originate in the left (distal) versus the right (proximal) region of the colon. The normal colonic tissues on the left and right sides originate from unique developmental origins [36]. Most CRCs (85%) originate on the distal side of the colon, are enriched in *KRAS*, *PIK3CA* and *TP53* mutations and display copy number variation (CNV) and chromosomal instability (CIN), in which key tumor suppressor genes are deleted and/or mutated. This contrasts with serrated CRCs located in the proximal colon, which are enriched in the *BRAF* mutation, are diploid (non-CIN) but are hypermutated, display microsatellite instability (MSI) due to *MLH1* silencing by DNA methylation and are more prevalent in older women. Interestingly, proximal tumors present with a more diverse panel of somatic mutations compared to distal (non-hypermutated) tumors, namely with enrichment for mutations in *ACVR2A*, *TGFBR2*, the mismatch repair genes *MSH3* and *MSH6* and the DNA polymerase *POLE* [31]. From a treatment perspective, patients with proximal tumors showed worse OS and recurrence-free survival (RFS) than patients with left-sided primary tumors after partial hepatectomy or tumor ablation to remove liver metastases [2].

The Cancer Genome Atlas (TCGA) presented whole genome sequencing (WGS) and whole exome sequencing (WES) on nearly 300 CRCs to show mutations in *APC*, *TP53*, *KRAS* and *PIK3CA* in left-sided tumors, while mutations in *ACVR2A*, *TGFBR2*, the mismatch repair genes *MSH3* and *MSH6* and the DNA polymerase *POLE* were enriched in right-sided tumors. Giannakis and colleagues performed WES on 619 archived CRC tumors and normal-adjacent tissue pairs to identify novel mutations in *BCL9L*, *CTCF*, *KLF5* and *RBM10*. In addition, the extent of somatic mutation correlates with the formation of neoantigen peptides that invoke anti-tumor immunity and is tied to lymphocytic infiltrations and patient survival [37].

### 3.1. Racial Differences in CRC Genetics

An analysis of genetic alterations in primary CRCs between African American and Caucasian patients showed elevated *KRAS* mutation rates in African American patients, as well as mutations of three genes (*EPHA6*, *FLCN* and *HTR1F)* only in African American CRC patients, although these mutations have a low frequency (3–6%) [8,38]. Guda and colleagues [38] performed next-generation sequencing of stage IV tumors with liver metastases from African American patients to identify mutations linked to metastatic disease. These data were analyzed to ultimately identify a panel of 15 genes that are more likely to be mutated in African Americans with CRC [38]. As a follow-up, Wang et al. then showed that stage I–III African American patients with these mutations were more prone to developing metastatic disease [39].

CNV analyses on primary CRCs from African American and Caucasian patients showed African American-specific aberrancies on chromosomes 11, 17p 20q and X [40,41,42], as well as elevated MSI. Moreover, an analysis of *TP53* alterations showed that while *TP53* mutation profiles are similar between African American and Caucasian CRC patients, the Pro/Pro polymorphism in codon 72 occurs at a higher frequency in African American patients and is linked to poor patient outcomes [43]. A comparison of somatic mutations among CRC patients showed that African American patients had a higher tumor mutation burden (TMB) than Caucasian patients in both early-onset and late-onset cases [44].

Early-onset African American patients also had increased *ATRX* mutations as well as significantly different mutation frequencies in *ATRX*, *APC*, *FBXW7*, *LRP1B*, *PIK3CA* and *RNF43* across racial groups, suggesting that somatic mutation profiles occur as a function of race within early-onset tumors [44]. In contrast, a report from Xicola et al. [45] used WES in early-onset CRCs to demonstrate that African American patients had reduced TMB, CNV and mutation frequencies in *APC* and other known CRC driver genes compared to Caucasians but harbored loss-of-function mutations in *BCL9L*, a negative regulator of B-catenin. Interestingly, Wnt regulatory genes were silenced by promoter DNA hypermethylation in these tumors, suggesting distinct molecular routes for CRC tumorigenesis (discussed in Section 4: DNA Methylation Alterations) [45].

### 3.2. Genetic Features of CRC Metastases

Since CRC mortality is driven by metastasis, examining the genetic profiles of CRC metastases may identify novel molecular alterations to identify the changes that occur during metastasis and those that can be exploited for patient treatment and surveillance. Vermaat performed targeted sequencing of over 1200 cancer-related genes across 21 primary CRC/liver metastasis pairs to show that liver metastases are genetically unique from those of the primary tumor, with 817 coding region alterations exclusively in the liver metastases [46]. This analysis identified the most common gene mutations (*KRAS*, *BRAF*, *PTEN*, *PIK3CA*, etc.) as well as other mutations in genes in the EGFR and TGF-B pathways that influence treatment success [46]. An analysis of CNV in primary tumors from 349 metastatic CRC patients from the CAIRO and CAIRO2 phase III clinical trials [47] identified gains at chromosomes 1q, 7, 8q, 13 and 20 and the most common losses at 1p, 4, 8p, 14, 15, 17p and 18. In addition, gains of chromosomes 8p, 11q, 12p, 13q, 18q, 19q, 20q and X were present in patients who developed metastases, and variations of 194 chromosomal regions were associated with progression-free survival (PFS) [47]. Ishaque et al. [48] performed WGS on colorectal adenomas, carcinomas and metastases and identified mutations in *AKT*, *BRCA2*, *FAT1*, *FGF1*, *KDR* and *NOTCH1* that may be therapeutic targets for disease treatment.

Yaeger et al. [49] analyzed sequencing data from 478 primary CRCs and 533 metastases from 979 patients with metastatic disease and 123 primary CRCs from patients with early-stage disease to find alterations related to metastasis. The mutation profiles of primary tumors and metastases were overall highly concordant. However, metastases harbored novel *APC* intron splicing alterations as well as *CTNNB1* deletions. As shown by other reports, patients with right-sided primary metastatic disease were of older age at the time of diagnosis and had reduced survival, increased mutation burden, and enrichment of *AKT1, KRAS*, *PIK3CA*, *PTEN*, *RNF43* and *SMAD4* mutations. In contrast, left-sided tumors exhibit the following characteristics: (1) display mutations in *APC*, *NRAS* and *TP53*; (2) display amplification enrichment in receptor tyrosine kinase signaling genes *ERBB2*, *ERBB3*, *EGFR* and *FGFR*; (3) do not display mutations or CNV in genes related to cell division; and (4) may be more prone to fluctuations in intestinal microbiomes, suggesting that patients with left- and right-sided CRCs have unique molecular pathways of metastasis. Comparing these data to a separate data cohort of metastatic tumors, as well as to The Cancer Genome Atlas (TCGA) and International Cancer Genome Consortia (ICGC) data for primary CRC mutations, Mendelaar et al. [50] identified the common mutations of left- and right-sided CRCs. Interestingly, the Mendelaar report also identified mutations in long non-coding RNA (lncRNA) genes, specifically noting that *LINC00672* mutations are associated with treatment response. In addition, the presence of *FBXW7* mutations signaled poor patient response to EGFR-targeted therapies.

## 4. DNA Methylation Alterations

Epigenetic alterations are hallmarks of CRC and every other tumor type. The main facets of epigenetics, DNA methylation, chromatin modifications of histone tail proteins, nucleosome occupancy and non-coding RNAs, namely microRNA (miRNA) and long non-coding RNA (lncRNA), are key for gene regulation and higher-order organization of chromatin. These epigenetic facets work in symphony to compose multiple levels of gene regulation. In human cancers, epigenetic aberrancies function to silence tumor suppressor genes and activate oncogenes. However, epigenetic marks are reversible via passive or active mechanisms, making them attractive therapeutic targets for disease treatment in reverting epigenome profiles to a more normal-like state. For the purposes of epigenetic alterations in CRC, we will focus on DNA methylation in this review.

DNA methylation in human and other mammalian tissues is generally defined as the addition of a methyl (-CH_3_) group to the C-5 position of cytosine nucleotides in the 5′-CG-3′ or CpG sequence context. Methyl marks are placed by the enzymatic activities of DNA methyltransferases (DNMTs) that use *S*-adenosylmethionine as a required co-factor. In human cells, DNMT1, DNMT3A and DNMT3B are largely responsible for DNA methylation catalysis, in which DNMT1 is involved in immediately copying the DNA methylation patterns from the parental strand to the nascent daughter strand during DNA replication, while DNMT3A and DNMT3B are responsible for placing methyl-marks at CpGs that were originally unmethylated. However, DNMT1 and DNMT3B are thought to work together to maintain DNA methylation profiles. DNMT3B also serves as an accessory scaffold protein in recruiting DNMT3A to specific CpG sites and is a key DNMT enzyme for catalyzing DNA methylation of gene body (transcribed) regions, in which DNMT3B is recognized by SETD2 for generating histone H3 lysine 36 trimethylation (H3K36me3) marks that are correlated with active gene expression.

While DNA methylation is pharmacologically reversible through the activities of DNMT inhibitors such as 5-azacytidine (Vidaza, AZA), 5-aza-2′-deoxycytidine (decitabine, DAC) and S110, enzymatic DNA demethylation is catalyzed in a stepwise manner through oxidative conversion of 5-methylcytosine (5mC) to 5-hydroxycytosince (5mC) by the TET family of enzymes. After TETs convert 5mC to 5hmC, 5hmC marks are further converted to 5-formylcytosine (5fC) and 5-carboxylcytosine (5caC), after which both 5fC and 5caC marks are ultimately replaced with unmethylated cytosines through base excision repair [51,52,53].

DNA methylation aberrancies are common in CRC and all other cancer types in which gene-specific promoter DNA hypermethylation is concomitant with global DNA hypomethylation of non-CpG rich regions and repeat sequences. Promoter DNA hypermethylation may correlate with gene silencing, while gene body DNA methylation and CpG poor DNA hypomethylation may be positively associated with gene expression. Epigenetic driver events in CRC tumorigenesis are limited and include epigenetic silencing of *MLH1*, *CDKN2A*, *MGMT*, *MLH1*, *RUNX3*, the *SFRP* gene family, *TPEF* and *VIM* (reviewed in [35]). More recently, Xu et al. [54] analyzed TCGA CRC CNV, DNA methylation and gene expression data to identify *LRRC26* and *REP15* as prognostic CRC driver events in which high *LRRC26* and *REP15* expression was inversely correlated with the presence of metastases and tumor stage.

Most cancer-specific DNA methylation changes do not result in gene expression changes but are biomarkers of disease. DNA methylation-based biomarkers are highly effective in disease monitoring and predicting patient outcome and response to treatment. DNA methylation biomarkers can be found in tumor-derived cell-free DNA (cfDNA) in patient blood serum or plasma, urine and fecal matter. DNA methylation markers include *ALX4*, *BMP3*, *CDH1*, *CDKN2A*, *MGMT*, *MLH1*, *NRDG4*, *PRIMA1*, SDC2, *SEPT9*, *SFPR2*, *TFPI2*, *TMEFF2* and *VIM* in blood, stool and/or urine specimens [55,56,57].

Two DNA methylation biomarker panels are FDA approved for early detection of CRC: (1) *SEPT9* in tissue and blood plasma, marketed as Epi proColon, and (2) *BMP3* and *NDRG4* DNA methylation with *KRAS* mutation screening in stool, marketed as Cologuard. It should be noted that *SDC2* DNA methylation testing in stool samples is approved in Korea (EarlyTect-Colon Cancer) and China (Colosafe) for CRC detection, and a yet unapproved test described as ColoDefense includes screening for both *SDC2* and *SEPT9* DNA methylation (reviewed in [58]). *SEPT9* DNA methylation in plasma shows high sensitivity (60–80%) and specificity (80–99%) in identifying stage I and II disease and outperforms both carcinoembryonic antigen (CEA) and fecal occult blood tests for CRC detection. However, *SEPT9* DNA methylation shows low sensitivity in detecting adenomas (summarized in [59]).

The large extent of DNA methylation changes in CRC has proven fruitful for the development of DNA methylation-based marker panels that predict patient outcome and response to treatment, as well as those that are linked to tumor stage and grade. Several reports have used DNA methylation data to build models related to patient outcome, which include the following: (1) Zhang and colleagues identified differential DNA methylation in *B3GNT7*, *CHN2*, *MUC12* and *TBX20* that is linked to gene expression and poor patient outcome [60]. (2) An analysis of Illumina Infinium DNA methylation array data from metastatic and non-metastatic CRCs revealed a 20-probe classifier with prognostic utility for non-metastatic cases [61]. This classifier included probes in *BRD4*, FBXL18, KCNQ1 and *MET*, among others, and the DNA methylation score of this panel was positively correlated with patient survival [61]. (3) Gong et al. mined TCGA CRC DNA methylation data to compile a three-gene (*NR1H2*, *SCRIB* and *UACA*) DNA methylation signature that correlated with patient OS irrespective of patient age and gender [62]. (4) Xie et al. [63] used tumor-derived cfDNA from blood plasma to evaluate the performance of a 14-gene panel (*ARHGEF4*, *AV3*, *BMP3*, *CHST2*, *DOCK10*, *IKZF1*, *LRRC4*, *NDRG4*, *OPLAH*, *PPP2R5C*, *PDGFD*, *QKI*, *SFMBT2* and *ZNF625*). The panel detected metastatic CRC with 90% sensitivity and 90% sensitivity and is superior to CEA measurements. (5) Deng and colleagues [64] analyzed TCGA CRC DNA methylation data to develop a 23-probe panel that is associated with disease progression. (6) Cai et al. [65] evaluated a panel of nearly 600 cancer-specific DNA methylation markers on CRCs and normal colonic tissues to compile a panel of 150 differentially methylated regions (DMRs) that were subsequently evaluated on 328 blood plasma samples. From this analysis, 24 DMRs were subject to validation across CRCs, normal-colon and blood plasma samples to arrive at six top markers: *BCAN*, *BCAT*, *IKZF1*, *VAV3* and two regions of *SEPT9* that were predictive of disease recurrence and outperformed CEA expression to detect CRCs across all four disease stages. (7) Muthamilselvan et al. [66] surveyed TCGA CRC DNA methylation data to identify markers of CRC specific for stage: stage-I (*FBN1*), stage-II (*FOXG1*), stage-III (*HCN1*) and stage-IV (*FAM123A*, *LAMA1*, *NELL1*, *ZNF135*). (8) Chen and colleagues also analyzed TCGA DNA methylation data to identify a panel of five immune-related genes—*FGF5*, *LTBP4*, *PIK3CD* and *PLXNC1* and *SCTR*—the composite DNA methylation signature of which correlated with prognosis for stage II and III CRC patients [67]. (9) As mentioned above, early-onset CRC incidence rates are rising [24,25], in contrast to lower incidence rates of late-onset disease seen in many parts of the developed world. DNA methylation profiles in early-onset CRCs display more extensive global DNA hypomethylation than late-onset cases [68], and a recent study from Joo and colleagues [69] used Illumina DNA methylation microarray profiling to identify 234 CpG regions with differential DNA methylation as well as accelerated cellular aging using epigenetic clocks (please see Section 4.3: Epigenetic Clocks) in early-onset cases.

### 4.1. CpG Island Methylator Phenotypes (CIMPs) in Colorectal Cancer

In addition to being used for predictive and prognostic purposes, cancer-specific DNA methylation profiles can be categorized to identify tumor subgroups. Indeed, Toyota and colleagues [70] first described a panel of cancer-specific DNA methylation events in a subgroup of CRC patients, termed CIMP. CIMP-specific loci remain unmethylated in non-CIMP tumors and normal-adjacent colonic mucosa, and patients with CIMP DNA methylation only represent 15–20% of all CRC cases. Weisenberger et al. [71] confirmed CIMP (now termed CIMP-high, CIMP-H) and developed a five-gene classification using real-time PCR technology: *CACNA1G*, *IGF2*, *NEUROG1*, *RUNX3* and *SOCS1*. CIMP is highly concordant with *BRAF* (*V600E*) point mutations, *MLH1* epigenetic silencing, MSI and a hypermutation profile but not *TP53* mutations or chromosomal instability, and it occurs in right-sided tumors and is enriched in female patients with poor clinical outcome [31,71,72]. Ogino et al. [73] subsequently identified the CIMP-low (CIMP-L) subgroup of CRCs that display reduced DNA methylation at CIMP-specific loci, are enriched with *KRAS* mutations, chromosomal instability and microsatellite stability and are present in the right side of the colon in male patients [31,72,73]. Depending on the analysis, unsupervised clustering of CRCs unveils three to four DNA methylation-based subgroups CIMP-H, CIMP-L and non-CIMP, in which non-CIMP tumors are categorized as either one or two subgroups [31,72,74,75].

Stratifying CRCs by CIMP status and other molecular co-variates has been shown to result in a profound dissemination of CRC patient outcome and response to therapy. In pooling data for over 5000 CRC patients, Phipps et al. [76] categorized CRCs into five groups based on their molecular profiles: (1) CIMP-positive, *BRAF* mutant (*BRAF*-mut), *KRAS* wild-type (*KRAS*-wt), high microsatellite instability (MSI-H); (2) CIMP-positive, *BRAF*-mut, *KRAS*-wt, microsatellite stable (MSS) or low microsatellite instability (MSI-low); (3) CIMP-negative, *BRAF*-wt, *KRAS*-mut, MSS/MSI-low; (4) CIMP-negative, *BRAF*-wt, *KRAS*-wt, MSS/MSI-low; and (5) CIMP-negative, *BRAF*-wt, *KRAS*-wt, MSS/MSI-low. Group 2 patients displayed the worst outcome, while MSI-H status was significantly associated with improved patient outcome, especially in cases harboring CIMP and *BRAF*-mut. In addition, Group 1 and Group 2 patients had significant associations with disease-specific survival (DSS), while Group 3 patients had the worst DSS, most notably in Group 3 patients with stage I-III disease. Indeed, *KRAS*-mut patients showed poor survival overall, specifically in MSS/MSI-low cases. These findings were also shown by Murcia and colleagues [77], who performed similar molecular analyses on 878 CRC patients, in that CIMP-positive, BRAF-mut and MSS patients had the worst prognosis. This analysis also showed that 5-fluorouracil (5-FU)-based therapies demonstrated improved prognosis in patients with either CIMP-negative, *KRAS*-mut and MSS status or CIMP-negative, *BRAF*-wt, *KRAS*-wt and MSS status.

Given these complicated aspects of CRC biology, a set of four consensus molecular subgroups (CMS) were developed that include RNA expression signatures [78]: CMS1 (MSI immune), CMS2 (canonical), CMS3 (metabolic) and CMS4 (mesenchymal) (Figure 3). CMS1 tumors display CIMP-H, MSI, *BRAF*-mut, DNA hypermutation, immune infiltration and activation and are linked to poor patient survival after relapse. CMS2 tumors are non-CIMP, with extensive CNV and activated WNT and MYC signaling. CMS3 tumors display CIMP-L DNA methylation, MSI/MSS, nearly diploid copy number, *KRAS*-mut and metabolic dysregulation. CMS4 tumors are non-CIMP but display CNVs, stromal cell infiltration, activated TGF-β and angiogenic signaling, and CMS4 patients show poor survival. To simplify CMS classification, CMS1 status can be effectively determined by measuring MSI status, CMS2 and CMS3 subgroups can be stratified by differential DNA methylation of specific loci, and CMS4 tumors can be determined by immunohistochemistry and gene expression classifiers [79,80,81,82].

### 4.2. DNA Methylation Profiles of Metastatic Disease

While molecular features of the primary tumor are expected to be represented in tumor metastases, there is an increasing body of evidence that metastases are characterized by novel molecular features that are not present in the corresponding primary tissue. Epigenetic profiling of liver metastases is limited compared to the primary tumor; however, several reports over the past decade have painted interesting portraits as to the role of DNA methylation changes in CRC. (1) Global DNA hypomethylation is common in CRC, in which non-CpG rich sequences in single copy loci and repetitive elements lose DNA methylation compared to normal tissues (reviewed in [35]). In comparing *LINE-1* repetitive element DNA methylation levels between primary CRCs and liver metastases, Hur et al. [83] detected significantly lower levels of *LINE-1* DNA methylation in metastases compared to primary CRCs, also resulting in the activation of proto-oncogenes, including *MET*, *CHRM3* and *RAB3IP*. (2) Ili et al. [84] compared primary CRCs to normal-adjacent tissues and lymph node metastases using genome-scale bisulfite sequencing and found that non-CpG island DNA hypomethylation and CpG island hypermethylation were extensive in primary tumors but attenuated in lymph node metastases. However, there are additional CpG islands that display DNA hypermethylation in lymph node metastases over primary tumors. Finally, an analysis of CpG islands that show concordant DNA methylation between primary tumors and lymph nodes included *BDNF*, *FIGN*, *HCN4*, *HTRA3* and *STAC2*, in which DNA methylation status of the panel was significantly associated with poor patient outcome. (3) Orjuela et al. [85] used methyl binding domain capture followed by sequencing to show that DNA hypermethylation of the primary tumor is well represented in liver metastases, although DNA hypomethylation was detected in the metastases compared to the primary tumors. (4) Yu and colleagues [86] used bisulfite sequencing to profile 30 CRC tumors and 19 lymph node metastases to ultimately identify DNA methylation of *LBX2* as a biomarker of metastasis that outperforms CEA and imaging testing. (5) An analysis of DNA methylation in 59 early- and late-stage CRCs, normal-adjacent tissues and metastases identified DNA methylation signatures of metastases in early-stage tumors that predicted outcome and the possibility of metastasis [87]. (6) DNA hypermethylation of *IMPA2* correlated with gene silencing in primary tumors and metastases and predicted poor patient outcome and advanced stage. (7) Bisulfite pyrosequencing of *CDH1*, *CDH13*, *CDKN2A (p16)*, *CDKN2A (p14)*, *ESR1*, *HPP1*, *MGMT*, *MINT1*, *MINT2*, *MINT31*, *MLH1*, *THBS1* and *TIMP3* across paired primary tumor-liver metastasis pairs, as well as CRCs with or without metastases, showed stochastic variability as a function of metastasis and suggest that cancer-specific DNA changes mostly occur prior to metastasis [88], which is in agreement with the other reports.

### 4.3. Epigenetic Clocks

Cancer is a disease of aging since increased age correlates with cancer risk, and age-related DNA methylation alterations have been described (reviewed in [89,90,91]). Although much less prevalent than cancer-associated DNA methylation alterations, global age-related DNA hypomethylation exists with regional DNA hypermethylation. In 1994, Issa and colleagues first identified age-related DNA hypermethylation of the human *ESR1* CpG island [92]. Subsequently, age-related DNA hypermethylation changes were identified in human tissues [93,94,95,96]. As technologies to measure DNA methylation across the genome improved, so did the ability to identify age-specific DNA methylation markers that measure organismal age [97]. Using Illumina DNA methylation array data, Horvath identified 353 CpG sites that represented an epigenetic clock after analyses of data from over 6000 samples [98]. The Horvath clock was compatible with a large array of tissue and cell types. Alternatively, the Hannum clock [99] surveyed a disparate set of 71 CpGs to calculate cellular age in blood. Two other epigenetic clocks were subsequently developed for use with blood samples that utilize 3–513 CpGs [91,100,101].

### 4.4. Epigenetics of Racial Disparities in CRC and Disease Risk

Epigenetic clocks have been important contributors to better understanding age-related diseases such as Alzheimer’s disease, human cancer and cardiovascular disease, as well as how epigenetic age is linked to inherited syndromes and other diseases [91]. Epigenetic clocks help to identify factors that accelerate or decelerate epigenetic age. Since cancer is a disease of aging, it is understood that epigenetic age acceleration occurs in colorectal cancer and other human cancers. Moreover, it is known that African Americans are diagnosed with advanced stage disease on the right side of the colon at an earlier age compared to Caucasians. Devall and colleagues [102] generated Illumina EPIC DNA methylation microarray data on matched left- and right-sided normal colon tissues from European Americans (Caucasians) and African Americans to identify if epigenetic age acceleration is correlated with CRC risk and sidedness using the Horvath model. Interestingly, African Americans displayed age acceleration in the right (proximal) side of the colon compared to Caucasians, while Caucasians displayed age acceleration on the left (distal) colon tissue specimens. In general, more African Americans showed older right-sided colons than left-sided colons compared to Caucasians. In addition, an analysis of differentially methylated regions (DMRs) between the right and left sides of the colon showed an enrichment for DNA hypermethylation in the right-sided colon in African Americans—these regions were linked to colonic enhancers as well as CRC tumorigenesis and aging. Notably, in normal rectal tissues, European Americans displayed increased epigenetic age acceleration compared to African Americans; race-specific, age-associated DMRs were identified, and epigenetic drift of age-related DMRs was more pronounced in African Americans [103]. Age-related DMRs were represented across both racial groups, suggesting that age acceleration is a driver of disease risk [103].

Indeed, DNA methylation changes in *ALU, LINE-1* and *SAT2* repetitive elements have been linked to environmental exposures as well as education, occupation, social status, wealth and family income status (reviewed in [104]). With this in mind, Konradsen and colleagues [105] performed a meta-analysis of 27 studies that involved stage III CRC patients with socioeconomic markers that include education, employment, financial status, income level and health insurance and showed that patients with low socioeconomic status had significantly increased time to start treatment and lower odds of receiving adjuvant chemotherapy. Finally, Alonso et al. [106] identified *MGMT* and *ADAMTS4* DNA hypermethylation in normal colonic mucosa, which are suggestive of an epigenetic field defect since these regions display DNA hypermethylation in CRC. Interestingly, DNA hypermethylation of these loci was more pronounced in the proximal colon in older African Americans and may serve as effective biomarkers for CRC risk in African Americans.

In contrast to cancer risk, there are only a handful of examples of CRC DNA methylation data that are linked to racial health disparities. A global meta-analysis of CIMP prevalence showed that it was highest in Africa and lowest in Asia, Europe and Australia [107]. In addition, a comparison of cancer-specific DNA methylation between African American and Caucasian CRC patients identified DNA hypermethylation of *ARHGEF4*, *CHL1*, *CHL4*, *GDF*, *ITGA4*, *NELL* and several microRNAs including *miR-9-3p* and *miR-124-3p* in African American CRC patients [108].

## 5. CRC Gene Expression Profiles

### 5.1. Gene Expression Subgroups and Functional Networks

Much like genetic (mutations and CNV) and epigenetic (DNA methylation) alterations, gene expression changes are common in CRC. From a tumor subgrouping perspective, CRC mRNA gene expression subgrouping generally follows the DNA methylation-based grouping with three subgroups: CIMP/MSI, invasive and chromosomal instability (CIN) [31]. In contrast, an unsupervised clustering analysis of mRNA gene expression data from 443 CRC patients revealed six subgroups: (C1) CIN with downregulated immune pathways, (C2) mismatch repair, (C3) *KRAS*-mut, (C4) cancer stem cell, (C5) CIN with Wnt pathway upregulation and (C6) CIN with a normal-like expression profile [109]. The C2 subgroup is linked to CIMP with *BRAF*-mut, but groups C2, C3 and C4 were enriched for CIMP and proximal primary tumor location. Alternatively, C1, C5 and C6 subgroups were enriched for CIN, non-CIMP and *TP53*-mut. The C4 subgroup was enriched for metastatic disease; however, there were no associations between subgroup and tumor staging. It should be noted that subgroups C4 and C6 were associated with shortened relapse-free survival.

Cancer-specific gene expression profiles are important in identifying functional networks important for cancer development and progression. Gene networks based on large expression data sets help to identify unique interactions within and between individual networks and expression pathways. Three recent examples examine this. First, Kim and colleagues analyzed CRC gene expression data sets [110] from unique regions of resected tumor tissues to extrapolate essential gene expression profiles in heterogeneous clinical tumor specimens and assign clinical specimens to specific subgroups. Notably, this analysis revealed that after stratifying the expression data by cancer specificity, residual expression data are present and unique for each patient and are cancer cell intrinsic. After clustering these residual gene expression profiles between patients, cancer cell-intrinsic subgroups are present and display novel prognostic value. Second, Emmert-Streib and colleagues [111] developed a gene regulatory network based on colon cancer gene expression data from the International Genomics Consortium with the goal of identifying co-regulated networks and the driver regions controlled by these networks at the chromosomal level to better understand CRC tumorigenesis. Of note, trans-chromosomal pairings are minor in CRC, with chromosome 22 genes mainly involved in these interactions. At a gene level, the main hub genes include *OR7E104P*, *GATA1*, *NPHP3-AS1*, *POM121L12*, *ZNF843*, *NYX*, *SLC22A25*, *C20orf203* and *SLC4A1*, and the enriched pathways are related to cell cycle, mitosis, M phase, membrane targeting and translation. Third, Su et al. [112] mined the TCGA CRC gene expression data to develop CRC biomarker networks for cancer diagnosis and staging. After first identifying cancer-specific gene expression profiles, the group ultimately identified genes associated with tumor stage that include *ADH1B*, *CLEC3B*, *GCNT2*, *GLDN*, *GLP2R*, *GREM2*, *JCHAIN*, *LYVE1*, *MAMDC2*, *PLP1*, *SCARA5* and *TMEM100*. In addition, individual expression levels of *SULT1B1*, *PTGDR2*, *GPR15*, *BMP5* and *CPT2* were significantly associated with OS. Pathway analyses showed enrichment of the curated expression panel for processes including metabolic processes and small molecule transport.

### 5.2. Gene Expression Specific to Race and Ethnicity

Based on whole transcriptome data, the Oncotype DX Colon Cancer Assay panel of 12 genes was developed to predict recurrence after resection of stage II/III disease [113,114,115,116]. The 12-gene panel consists of *BGN*, *FAP*, *GADD45B*, *INHBA*, *MK167*, *MYBL2*, *MYC* and five reference normalization genes [113]. Interestingly, a comparison of the 12-gene panel between Caucasians and African Americans with stage II CRC did not show any differences, nor were there any expression differences for each individual gene between the two groups [113]. These data indicate that although there are health disparities between Caucasian and African American CRC patients, they are not represented in this gene expression panel. However, comparison of African American and European American CRC disease using Agilent gene expression microarrays did reveal 95 genes with expression differences between the two groups [117]. This was reduced to a panel of 10 genes that can accurately predict ethnicity: *ADAL*, *ANKRD36B*, *ARHGAP6*, *CRYBB2*, *C17orf81 PSPH*, *TRNT1*, *VSIG10L*, *WDR8* and *ZNF835* [117].

### 5.3. Gene Expression Related to CRC Metastasis

Gene expression data can also be used in profiling CRC metastases. A meta-analysis of gene expression data sets of primary CRCs and liver metastases [118] unveiled a panel of 35 differentially expressed genes (DEGs), with the most dysregulated genes including *AHSG*, *APOA2*, *CP*, *FGA*, *F2*, *HPX*, *HRG*, *ITIH2*, *PLG* and *SERPINC1*. Among these, expression of ITIH2 was statistically significantly associated with survival, suggesting that it is important for CRC metastasis [118]. In comparison, Wu et al. examined whole transcriptome RNA expression (RNA-seq) and whole exome sequencing (WES) to examine gene expression and mutations/CNV, respectively, across 99 stage IV patients who underwent resection for only the primary tumor to identify correlates with stage IV disease [119]. Interestingly, among the clinical co-variates of tumor stage, gender, race, tumor location and tumor pathology, only tumor location was a significant determinant of outcome. In addition, mutations of key genes including *APC*, *TP53*, *KRAS* and *PIK3CA* did not correlate with patient outcome. The gene expression analyses, however, identified 97 genes with aberrant expression, including significant differences in *DSCAML1*, *FGF18*, *NEUROD1*, *PLAC1*, *SAA2*, *SFTA2* and *OTOP3* that were associated with patient outcome. *FGF18* and *DSCAML1* expression had a protective effect, while the expression of the other genes in the panel was positively associated with outcome risk [119].

## 6. Influence of the Tumor Microenvironment and Gut Microbiome on CRC Molecular Profiles

Increasing amounts of evidence indicate that the tumor microenvironment (TME) is a substantial contributor to CRC tumorigenesis and disease progression. The TME is composed of several cell types surrounding the colorectum, including adipocytes, endothelial cells, fibroblasts, immune cells and inflammatory cells [120,121]. Tumor cells and the TME communicate such that CRC tumor cells induce cytokine and growth factor production from the TME cell types to supplement tumor cell proliferation and immune escape and ultimately metastasis. [120,121]. From a molecular perspective in CRC, Wnt signaling is aberrant across 90% of all CRCs [31], and activated Wnt signaling correlates with the lack of T-cell infiltration [122]. In addition, CIMP-H tumors (CMS1, immune) with MSI-H display mutations, deletions and *AXIN2* super-enhancer DNA hypomethylation in immune-modulating and antigen-presenting genes to provide MSI-H CIMP tumors with an advantage of immune escape [122].

It is becoming clear that immune-related gene expression differences play a role in CRC health disparities. For instance, African Americans display reduced antitumor cytotoxic immunity [123]. Second, gene expression differences in inflammatory and immune response pathways are present between African Americans and Caucasians [117]. An analysis of TCGA CRC data by Curran and colleagues [124] revealed that African American CRCs express reduced levels of CD8+ T cells and macrophages but overexpress B cells in comparison to European Americans, as was described by Basa et al. [125]. Gene expression profiling also showed that major histocompatibility complex genes as well as immune-inhibiting and immune-stimulating genes were differentially expressed between the two groups of patients [124]. Overall, these findings indicate that B and T cell expression difference may help address the health disparities in CRC patients.

The gut microbiome consists of microorganisms required for digestive health and is also influential in CRC progression [126], indicating crosstalk between the microbiome and the host colonic epithelium (reviewed in [127]). The microbiome is plentiful in CRC and hepatocellular tumors [128,129], and reports have highlighted the role of the microbiome in regulating cancer treatment response [130,131,132]. Poore and colleagues analyzed TCGA WGS and RNA-sequencing data from tissue and blood across all 33 cancer types profiled and found microbiomes present in most forms of human cancer [133]. Using these findings, Uriarte-Navarrete et al. [134] stratified CRC data into early- and late-stage groups and measured the interactions of each microorganism with each human gene. Interestingly, they found that due to how gene-microbiome expression networks are organized, early-stage CRC involves instituting structural features of the tumor cells, whereas late-stage disease displays advanced metabolic features and signaling [134]. Another report in which CRC mRNA expression data was integrated with microbiome information from several datasets revealed that 41 overexpressed genes in CRC (including *AURKA*, *AURKB*, *BUB1*, *CCNA2*, *CDK1*, *CENPF*, *MAD2L1*, *TPX2* and *UBE2C*) are enriched in microtubule and tubule binding and were negatively associated with the presence of *fusobacteria* and positively associated with *blautia*, suggesting that these bacteria are interacting partners for modulating gene expression [135].

## 7. Drug Treatments and Therapeutic Strategies

Therapies using 5-fluorouracil (5-FU) have been the backbone of CRC treatments for several decades. Used as an inhibitor of DNA synthesis by preventing thymidine production, 5-FU has demonstrated efficacy in treating CRC patients and is now generally combined with (1) folinic acid (leucovorin) and oxaliplatin to form the FOLFOX chemotherapy regimen; (2) folinic acid and irinotecan to form the FOLFIRI regimen; (3) folinic acid, oxaliplatin and irinotecan in the FOXFOXIRI regimen [136]; or (4) XELOX, the combination of capecitabine and oxaliplatin.

Folinic acid is a vitamin B derivative that exacerbates the 5-FU cytotoxicity, irinotecan is a topoisomerase inhibitor that prevents DNA uncoiling and replication, and oxaliplatin is a platinum-based chemotherapy agent that inhibits DNA synthesis and DNA repair. Capecitabine (Xeloda) is an oral fluoropyrimidine that enters the cell and is converted to 5-FU in three steps [137], in which capecitabine is first converted to 5′deoxy-5-fluorocytidine by hepatic carboxylesterase, which is converted to 5′-deoxy-5-fluorouridine by cytidine deaminase. The substance 5′-deoxy-5-fluorouridine is converted to 5-FU by thymidine phosphorylase. Importantly, thymidine phosphorylase displays substantially higher activity in tumor tissues compared to normal tissues, thereby improving delivery of 5-FU specifically to tumor cells.

Most targeted CRC treatment strategies have focused on the epidermal growth factor receptor (EGFR) and vascular endothelial growth factor receptor (VEGFR) signaling pathways (summarized in [35]). The activation cascade of EGFR/KRAS/BRAF results in MEK1, MEK2 and ERK activation. ERK signaling activates oncogenic transcription factors that include ELK1, FOS, JUN and MYC. As a result, these transcription factors activate driver genes involved in cell proliferation, cell cycle progression and differentiation (reviewed in [35]). The VEGFR pathway also involves RAS and BRAF signaling but independently stimulates the PI3K/AKT/mTOR signaling cascade to promote cell growth, differentiation and angiogenesis.

Several monoclonal antibodies, such as cetuximab and panitumumab, have been developed that target the EGFR pathway by binding to the EGFR extracellular domain to block the binding of EGF ligands [138,139]. Two phase III clinical trials [140,141] showed that the addition of cetuximab to FOLFOX or FOLFIRI in first-line treatment of *RAS*-wt metastatic CRC patients resulted in improved patient OS and PFS. However, these agents are only effective in *RAS*-wt tumors. In fact, only *RAS*-mutation status and, to a lesser extent, *BRAF*-mutation status, guide CRC therapeutic decisions [142,143], Approximately 40% of RAS-wt tumors are resistant to cetuximab after earlier-line treatment failure with irinotecan-based regimens [142].

Other agents effective in treating CRC patients include nivolumab, a monoclonal antibody that targets the PD-1 receptor of lymphocytes. Ipilimumab is a monoclonal antibody that inhibits CTLA-4 and subsequently activates the immune response, while pembrolizumab is a monoclonal antibody that also targets PD-1 and is FDA approved for mismatch repair-deficient metastatic CRC [144]. Pertruzumab and trastuzumab are monoclonal antibodies that target ERBB2 in ERBB2-positive CRC and are widely used in breast cancer treatment [145]. *ERBB2* amplification occurs in 3% of all metastatic CRCs and in 5% of CRC patients with *RAS*-wt tumors [146], thus representing an appreciably substantial patient subgroup that may benefit from ERBB2 inhibition. Bevacizumab is a monoclonal antibody that is directed towards the VEGF signaling pathway that is key for angiogenesis. Namely, bevacizumab inhibits tumor blood vessel growth and blood vessel recurrence and reverses permeability of the surviving vasculature. As a result, bevacizumab has been shown to be effective in improving CRC patient survival and is frequently used with 5-FU and other chemotherapies (reviewed in [147]).

Entrectinib is a small-molecule competitive inhibitor of tropomyosin neurotropic tyrosine receptor kinases A, B and C (TRKA, TRKB, TRKC), the proto-oncogene tyrosine protein kinase ROS1 and anaplastic lymphoma kinase (ALK) [148]. Similarly, larotrectinib is an inhibitor of TRKA, TRKB and TRKC. Specifically, TRKA activation induces phosphorylation of specific tyrosines across a series of proteins resulting in activation of multiple pathways that include PI3K/AKT and Ras [149,150,151]. TRKA is encoded by the neurotrophic tyrosine kinase receptor 1 (*NTRK1*) gene, and *NTRK1* alterations have been observed in several human cancers, including colorectal cancer. Interestingly, *NTRK1* is involved in multiple chromosomal rearrangements in human cancers that have oncogenic activity, including TPM3-TRK in colorectal cancer as well as MPRIP-NRTK1 and CD74 in non-small cell lung cancer [149]. The CRC TPM3-TRK fusion occurs at low frequencies (1–2%) in CRC [149] and co-occurs with *APC* and *TP53* alterations as well as DNA hypermutation, MSI-H, enrichment in *POLE*/*POLD1* mutations and location on the right side of the colorectum [152]. NTRK-altered CRCs are sensitive to both entrectinib and larotrectinib [68,153], suggesting that this subgroup of CRC patients may benefit from this personalized medicine approach.

### 7.1. Treatment Outcomes as a Function of Tumor Location and CIMP Status

CIMP-H tumors are generally found in the proximal (right) region of the colon; therefore, tumor location is an important determinant of CRC patient outcome (summarized in [35]). Even left and right colonic tissues have unique embryonic origins—right-sided colonic epithelial cells are derived from the midgut, while left-sided normal colonic mucosa is derived from the hindgut. In addition, colon crypt stem cell populations and embryonic gene expression differences begin in normal colonic tissues [154,155,156]. Left- and right-sided colon cancers display extensive differences in terms of genetic, epigenetic and transcriptomic profiles [154] as well as in clinical trials. A clinical trial of *KRAS*-wt metastatic colon cancer patients who were treated with either FOLFIRI or FOLFOX before treatment with bevacizumab or cetuximab showed significant differences in OS and PFS after stratification by tumor location [157,158], even though OS rates did not differ as a function of treatment arm. *RAS*-wt patients with left-sided tumors had improved PFS and OS compared to patients with right-sided tumors. Patients with left-sided *RAS*-wt tumors showed a significantly improved outcome when given cetuximab + FOLFIRI compared to right-sided, *RAS*-wt CRC patients [159]. Recently, Rossini et al. [160] performed a meta-analysis of five completed clinical trials (CAIRO5, FIRE-3, PARADIGM, PEAK and SWOG 80405) spanning over 2700 CRC patients to show that left-sided metastatic CRC patients showed improved OS after receiving anti-EGFR therapies compared to bevacizumab, while right-sided patients who received bevacizumab had longer progression-free survival but not significantly improved OS. Interestingly, there were significant correlations between tumor sidedness and treatment type with respect to PFS and OS, confirming the observations from other clinical trials.

From a molecular subtype perspective, CIMP-positive CRC patients have been evaluated in several clinical trials (summarized in [35,161]). In general, patients with right-sided tumors exhibit poorer OS and PFS and higher mortality rates than patients with left-sided tumors [162]; however, Min et al. and Van Rijnsoever et al. reported a survival benefit in stage II/III CIMP-positive patients when given 5-FU therapy [161,163,164]. CIMP-H CRC patients display worse DFS and OS than non-CIMP patients with microsatellite stable (MSS) or instable (MSI) status [165]. Similar findings were revealed after treatment with 5-FU in the FOLFOX setting, in which CIMP-positive patients generally showed no significant difference in DFS; however, OS rates improved. It should be noted that Zhang and colleagues did detect increased PFS and OS when CIMP-positive, metastatic CRC patients were given irinotecan followed by FOLFOX or when CIMP-positive patients received irinotecan, 5-FU and folinic acid [161,166,167]. CIMP-positive, *KRAS*-wt CRC patients who were given cetuximab demonstrated a shortened PFS compared to non-CIMP patients (reviewed in [161]). Finally, stratifying metastatic CRC patients by CMS gene expression subgrouping in the FIRE-3 clinical trial in which *KRAS*-wt patients received FOLFIRI + cetuximab or bevacizumab showed that CMS classification was significantly prognostic for PFS and OS [168]. Interestingly, CMS3 (right-sided, CIMP-L) and CMS4 (left-sided, non-CIMP) patients showed improved OS when given cetuximab compared to bevacizumab [168].

CIMP-positive tumors are highly enriched for the BRAF V600E point mutation and are resistant to EGFR-based therapies. In addition, treatments with BRAF inhibitors, including vemurafenib, as single agents have not demonstrated clinical efficacy [169,170]. Interestingly, clinical trials involving the combination of BRAF- and EGFR-inhibitors (vemurafenib and cetuximab) have shown improved performance for *BRAF*-mut CRCs [169,171,172,173]. Dabrafenib, another BRAF inhibitor, also showed an unexceptional response in treating BRAF-mut CRCs when combined with the MEK inhibitor trametinib, and response rates improved when dabrafenib (BRAF), trametinib (MEK) and panitumumab (EGFR) were combined. Similar findings were also found when combining vemurafenib (BRAF), cetuximab (EGFR) and irinotecan [169,174] or the combination of encorafenib (BRAF), cetuximab (EGFR) and binimetinib (MEK) (reviewed in [175]), supporting the observation that blocking EGFR signaling improves patient outcomes [169,176].

Clinical trials have also evaluated the efficacy of immune checkpoint inhibitors in treating *BRAF*-mut CRC patients. *BRAF*-mut and *BRAF*-wt CRC patients with microsatellite instability (MSI-H) given the anti-PD-1 antibody pembrolizumab showed improved outcomes [144,169]. Treatment-refractory, *BRAF*-mut, MSI-H CRC patients who were administered nivolumab (anti-PD-1) and ipilimumab (anti-T lymphocyte antigen-4) showed overall response rates similar to those for *BRAF*-wt CRC patients [169,177], although the duration of treatment response was shorter for *BRAF*-mut patients than for *BRAF*-wt patients. Finally, treatment of microsatellite-instable CRC patients with pembrolizumab (anti-PD-1) showed improved PFS compared to standard chemotherapy as a first-line approach [169,178].

### 7.2. Epigenetic Therapy in Colorectal Cancer

As described above, DNA methylation changes are pervasive in CRC and all other forms of human cancer and have roles in gene silencing and activation. DNA methylation is a dynamic and pharmacologically reversible epigenetic mark, making it an attractive therapeutic target summarized in [35]. Indeed, the global loss of DNA methylation in cancer cells, whether through genetic ablation or pharmacological intervention, results in dramatic cancer cell growth inhibition [179,180,181,182,183,184]. DNA methylation inhibitors, such as 5-aza-2′-deoxycytidine (decitabine, DAC), have anti-neoplastic effects on cancer cells through the reactivation of tumor suppressor genes [185,186] and downregulation of oncogenes [179]. DAC treatment also induces viral mimicry to stimulate an immune response and convert non-immune responsive tumors and/or those without T-cell infiltration (immune cold) to immune responsive tumors (immune hot) [180,181,182,184], which subsequently show increased responsiveness to chemotherapies [187]. This conversion starts with transcriptional reactivation of transposable elements (TEs) that include long terminal repeats (LTRs), short interspersed nuclear elements (SINEs) and long interspersed nuclear elements (LINEs) [180,181,182,183,184]. Therefore, DNA methylation inhibition combined with chemotherapeutic agents holds promise for cancer patient therapy.

### 7.3. Treatment of Metastatic CRC

Metastatic CRC represents a substantial therapeutic challenge, especially since the survival rates for patients with metastatic disease are dismal. Biller and Schrag [188] performed a meta-analysis of data from 222 clinical trials between 2014–2020 to better understand how molecular profiles affect treatment outcomes. Patients with *BRAF*-wt, *KRAS*-wt, *NRAS*-wt and metastatic CRCs have a median survival with treatment of 30 months [188]. However, stratifying patients by primary tumor location showed that right-sided patient median survival was 19 months compared to 34 months for patients with left-sided tumors [188]. These *BRAF*-wt, *RAS*-wt patients also show modest survival improvement of 2–4 additional months when given cetuximab and panitumumab with backbone chemotherapy compared to chemotherapy alone [188]. Left-sided metastatic CRC patients also demonstrated benefit from EGFR inhibition added to conventional chemotherapy when compared to bevacizumab and chemotherapy [188].

*BRAF*-mut patients saw a survival period of 9.3 months when given BRAF- and EGFR-based inhibitors compared to 5.9 months for those given standard chemotherapy [188]. Finally, patients with MSI or defective mismatch repair showed tremendous improvement in OS when given immunotherapy [188]. Patients with recurrent metastatic disease represent an especially challenging group, as there are no approved therapies that increase survival by more than three months, although treatment with regorafenib (a tyrosine kinase inhibitor), trifluridine (a thymidine analog) and tipiracil (a thymidine phosphorylase inhibitor) are FDA approved for third-line therapies [189] and show survival increases of 1–2 months [188].

## 8. Conclusions and Future Directions

In summary, CRC is a worldwide health burden, and CRC outcomes encapsulate socio-economic and racial disparities that affect patient diagnosis, treatment and outcome. From a biological standpoint, CRC is a heterogeneous disease in which biological covariates such as CIMP, *KRAS*/*BRAF* mutation, MSI and tumor location complicate treatment response and patient outcomes. CRC treatments historically target DNA synthesis and EGFR/VEGFR signaling; however, new treatment schemes have combined these with immunotherapies. Multi-omics profiling has provided new windows into CRC tumorigenesis and metastasis, but profiling minority populations with adenomas or adenocarcinomas is important for reducing health disparities and improving patient outcomes.

## Figures and Tables

**Figure 1 cancers-15-02934-f001:**
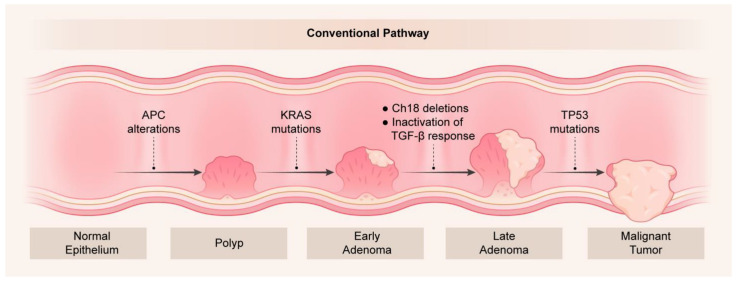
Genetic changes as a function of colorectal tumorigenesis. Key molecular alterations are shown to occur with progression from normal epithelium to polyps and tumors in the adenoma-carcinoma sequence originally described by Vogelstein [29].

**Figure 2 cancers-15-02934-f002:**
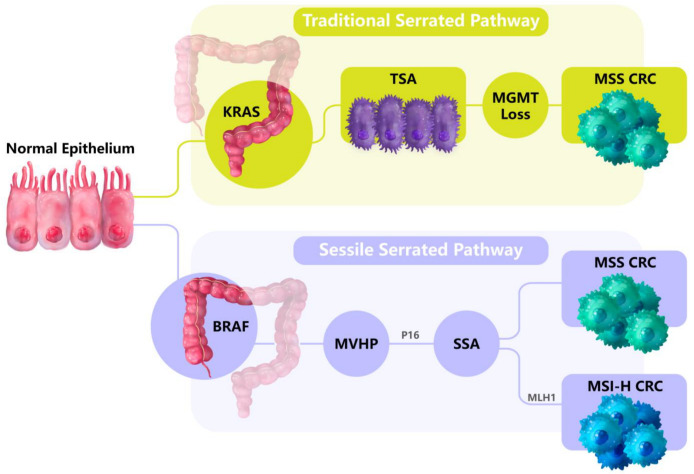
The serrated pathways of CRC tumorigenesis. Traditional serrated pathways develop in the distal (left) side of the colorectum and are linked to *KRAS* mutations. Alternatively, sessile serrated pathways develop in the proximal (right) side of the colorectum and are enriched in the *BRAF (V600E)* point mutation. From the normal colon tissues, microvesicular hyperplastic polyps (MVHPs) form that are the precursors to sessile serrated adenomas (SSAs). SSAs then develop into CRCs, and display microsatellite instability (MSI) if *MLH1* is epigenetically silenced.

**Figure 3 cancers-15-02934-f003:**
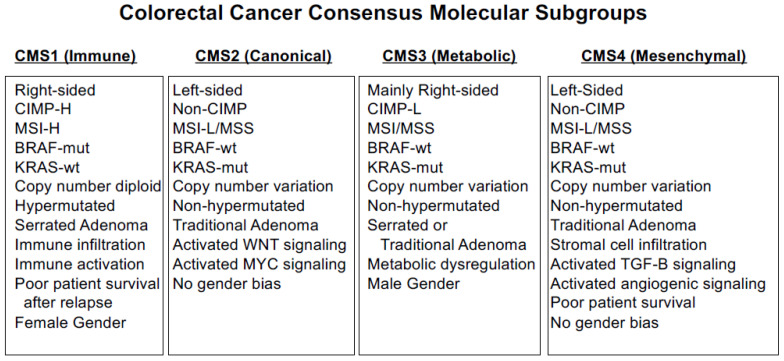
CRC Consensus Molecular Subgroups.

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
