# Peer review of "A Multi-Omics Overview of Colorectal Cancer to Address Mechanisms of Disease, Metastasis, Patient Disparities and Outcomes"

_cancers, 2023, doi:10.3390/cancers15112934_

Round 1

Reviewer 1 Report

The review by Yang and colleagues covers a broad range of omics-related topics related to colorectal cancer, including aspects of racial disparities, genetic, DNA methylation, and transcriptional changes linked to CRC, aspects of CRC metastasis, factors influencing treatment outcome, and finally some discussion of the microbiome and TME in CRC biology. The manuscript is overall well written and easy to read. The figures are helpful, although 1-2 additional figures and/or tables summarizing some of the epigenetic and transcriptional changes would make for easier reading. The authors even touch on epigenetic aging in CRC. While some discussion of immunotherapy is provided, the review might benefit from some additional discussion of how epigenetic marks and epigenetic inhibitors are being linked to immunotherapy outcomes (possibly improving these treatments). Overall these issues are minor and the review should become an important resource for those in the field once published.

minor corrections only

Reviewer 2 Report

The review by G.Yang et al. analyzed an important problem in cancer biology and practice, that is, the molecular 'portraits' of colon cancer in various population cohorts. The study is very well organized, detailed, thoughtful and profound. I recommend to accept the study in the present form. 

Author Response

We thank this reviewer for his/her positive and supportive evaluation of this manuscript.

Reviewer 3 Report

The main missing point is that the authors did not describe how they selected papers for this review.

Is chapter 7 wtritten for clinicians ? Chapter 7 is just mentioning a number of drugs and do not discriminate between early and metastatic disease. In addition they do not take sided-ness into account, sided-ness is very important for selection of targeted therapy. I recommend to skip chapter 7 or completely rewrite.

Line 634

“Upon entering the cell, capecitabine is converted to 5-FU to inhibit thymidylate synthase.”

It is a little more complicated than that. “Capecitabine is first metabolized in the liver to 5-deoxy-fluorocytidine by hepatic carboxylesterase and subsequently converted to 5-DFUR by cytidine deaminase, principally located in the liver and tumor tissues. Further catalytic activation of 5-DFUR to 5-FU then occurs at the tumor site by the tumor-associated angiogenic factor thymidine phosphorylase, thereby minimizing the exposure of healthy body tissues to action.systemic 5-FU.” (From van Cutsem et al, JCO 2000).

5FU has different ways of action not only inhibition of TS.

Line 646

“In fact, only RAS-mutation status guides CRC therapeutic decisions [138] but 40-60% of RAS-wt tumors are resistant to EGFR-based treatments.”

What about MMR, HER2, KRASG12C, BRAF and so on ? What about sided-ness ?

Please include references showing that 40-60% RAS-wt tumors are resistant to EGFR-based treatments.

Line 649

“Agents that do not specifically target the EGFR pathway include”

Why only these 4. What about bevacizumab, entrectenib and so on.

Line 672

“Patients with BRAF-wt, KRAS-wt, NRAS-wt and metastatic CRCs show modest survival improvement of 2-4 months when given cetuximab and panitumumab with backbone chemotherapy as compared to chemotherapy alone.”

Yes but the same accounts for oxaliplatin and irinotecan but the sequential use of combination therapy has improved OS from 6 months (before chemo) to more than 30 months. In addition, the authors do not mention that patients with BRAF-wt and RAS-wt are further selected by sidedness. In left-sided mCRC a combination of chemo with EGFR-inhibitor prolong median OS with 6-12 months compated to chemo + bevacizumab

Line 675

“In addition, BRAF-mut patients saw a similar survival improvement when given BRAF- and EGFR-based inhibitors compared to those given standard chemotherapy.”

Reference ?
